# Feasibility of EBUS-TBNA for histopathological and molecular diagnostics of NSCLC—A retrospective single-center experience

**Marija Karadzovska-Kotevska**[1,2]*, **Hans Brunnström**[3,4], **Jaroslaw Kosieradzki**[1], **Lars Ek**[1], **Christel Estberg**[1], **Johan Staaf**[2], **Stefan Barath**[1], **Maria Planck**[1,2]

**1** Department of Respiratory Diseases and Allergology, Skåne University Hospital Lund, Lund, Sweden, **2** Division of Oncology, Department of Clinical Sciences Lund, Lund University, Medicon Village, Lund, Sweden, **3** Division of Laboratory Medicine, Department of Genetics and Pathology, Region Skåne, Lund, Sweden, **4** Division of Pathology, Department of Clinical Sciences Lund, Lund University, Lund, Sweden

* Marija.Kotevska@med.lu.se

## Abstract

Endobronchial ultrasound-guided transbronchial needle aspiration (EBUS-TBNA) is a minimally invasive bronchoscopic procedure, well established as a diagnostic modality of first choice for diagnosis and staging of non-small cell lung cancer (NSCLC). The therapeutic decisions for advanced NSCLC require comprehensive profiling of actionable mutations, which is currently considered to be an essential part of the diagnostic process. The purpose of this study was to evaluate the utility of EBUS-TBNA cytology specimen for histological subtyping, molecular profiling of NSCLC by massive parallel sequencing (MPS), as well as for PD-L1 analysis. A retrospective review of 806 EBUS bronchoscopies was performed, resulting in a cohort of 132 consecutive patients with EBUS-TBNA specimens showing NSCLC cells in lymph nodes. Data on patient demographics, radiology features of the suspected tumor and mediastinal engagement, lymph nodes sampled, the histopathological subtype of NSCLC, and performed molecular analysis were collected. The EBUS-TBNA specimen proved sufficient for subtyping NSCLC in 83% and analysis of treatment predictive biomarkers in 77% (MPS in 53%). The adequacy of the EBUS-TBNA specimen was 69% for *EGFR* gene mutation analysis, 49% for analysis of *ALK* rearrangement, 36% for *ROS1* rearrangement, and 33% for analysis of PD-L1. The findings of our study confirm that EBUS-TBNA cytology aspirate is appropriate for diagnosis and subtyping of NSCLC and largely also for treatment predictive molecular testing, although more data is needed on the utility of EBUS cytology specimen for MPS and PD-L1 analysis.

## Introduction

Lung cancer, a severe disease with increasing incidence, is the leading cause of cancer-related death globally [1]. Investigation, diagnosis, precise staging, and genomic profiling of lung cancer is a demanding but fundamental process for adequate cancer treatment. For non-small cell lung cancer (NSCLC), representing 85% of all lung cancer cases, several new treatment

**Data Availability Statement:** All relevant data are within the paper and its Supporting information files.

**Funding:** MP (Maria Planck) received grants from Swedish Cancer society (190473PjO1H), Mrs. Berta Kamprads Foundation (FBKS-2020-7), Crafoord's Foundation (20209975), and Sjöberg's Foundation (2019-2011). The funders did not play any role in the study design, data collection and analysis, decision to publish or preparation of the manuscript. https://www.cancerfonden.se https://www.frubertakampradsstiftelse.se https://www.crafoord.se https://sjobergstiftelsen.se.

**Competing interests:** The authors have declared that no competing interests exist.

alternatives have emerged during recent years and may now include, for example, tyrosine kinase inhibitors, targeting tumor-specific mutations/fusion genes, or inhibitors of the immune checkpoint molecules PD1/PD-L1 (programmed death-ligand 1). Today's therapy decisions thus require thorough molecular analyses to identify clinically relevant alterations. In line with this, the Clinical Practice Guidelines for NSCLC of National Comprehensive Cancer Network (NCCN) 2017 recommend parallel diagnosis, staging, and molecular genetic testing [2].

Being proven as a minimally invasive and effective technique for assessment of the pathology of mediastinal and hilar lymph nodes (LN) along with pulmonary masses proximate to the airway, endobronchial ultrasound-guided transbronchial needle aspiration (EBUS-TBNA) has become a preferred investigation method amongst practitioners of pulmonary medicine worldwide. Performed in outpatient settings with high accuracy and minimal complication rate, EBUS-TBNA has been confirmed to be a safe procedure [3, 4]. EBUS-TBNA has now been accepted as a procedure of choice to diagnose and stage locally advanced lung cancer and is recommended by national and international guidelines [5, 6]. Supplementing EBUS-TBNA with transesophageal bronchoscopic ultrasound-guided fine needle aspiration (EUS/EUS-B-FNA) extends the ability to sample multiple intrathoracic LN stations as well as distant metastasis and structures below the diaphragm. Current American College of Chest Physicians (ACCP) lung cancer guidelines recommend EBUS-TBNA and EUS/EUS-B-FNA over invasive mediastinal and surgical staging as the initial staging of NSCLC [7].

In the majority of lung cancer cases, the diagnosis is confirmed by small cytological and/or biopsy specimens, with EBUS-TBNA as a frequent modality of tumor cell acquisition. Studies evaluating EBUS-TBNA in tissue sampling for histopathological diagnostics of lung cancer have demonstrated success rates ranging from 89% to 98% [8–11]. Molecular analysis can be performed on cytology or biopsy specimens acquired by EBUS-TBNA [12]. The adequacy of EBUS-TBNA samples for molecular testing depends primarily on the absolute number of viable tumor cells, the percentage of tumor cells in the material, and the sensitivity of the particular molecular test [13, 14]. Surveys from different centers demonstrate significant variations of EBUS-TBNA regarding diagnostic yield, sensitivity, negative predictive value, and success rate of genetic testing [11]. Many studies conclude that EBUS-TBNA specimens are sufficient for histological subtyping of NSCLC as well as targeted *EGFR* mutation and *ALK* gene fusion analysis [8–11].

A limited number of studies have evaluated the suitability of EBUS-TBNA specimen for parallel multiple gene alteration analysis [15–19]. Molecular profiling of lung cancer is today widely performed by massive parallel sequencing (MPS) using targeted gene panels. However, the adequacy of the EBUS-TBNA cytology specimen for analysis of predictive biomarkers using targeted MPS panels is insufficiently explored [11, 19]. This study aims to evaluate the adequacy and sufficiency of EBUS-TBNA cytology specimen for subtyping, molecular genetic profiling, and analysis of PD-L1 in a consecutive series of patients diagnosed with NSCLC.

## Materials and methods

### Analytical data

A retrospective review of the medical records of all patients examined with flexible and EBUS bronchoscopy as a part of the diagnostic process for suspected or known lung cancer, between January 1, 2017, and April 23, 2018, at the unit for Interventional Pulmonology (IP) at the Department of Respiratory Diseases and Allergology, University Hospital of Skåne, Sweden was performed. The University Hospital of Skåne is providing highly specialized patient care, diagnosis, and treatment for approximately 1.3 million people throughout the southern part of

Sweden. Bronchoscopy with EBUS-TBNA was introduced in the IP center in 2004. The IP center is one of Sweden's largest in volume, with more than 2000 interventions per year: approx. 800 EBUS-TBNA, 300 electromagnetic navigational bronchoscopies, over 100 advanced rigid bronchoscopies with endobronchial therapy and many chest tube insertions, thoracenteses, and medical thoracoscopies.

The Regional Ethical Review Board at the University of Lund, Sweden, approved this study (reg nr 2018/730). The Ethical Review Board permitted Marija Karadzovska-Kotevska to extract patients' data from the medical records. The data set was fully anonymized for all researchers, including Marija Karadzovska-Kotevska, before the initiation of the statistical analysis of the material. Patients' medical records were accessed between January 2019 and April 2019. Patients included in our study were examined with bronchoscopy with EBUS-TBNA for suspected or known lung cancer at the Unit for Interventional Pulmonology at University Hospital of Skåne in the period between January 2017 and April 2018. The Ethical Review Board did not request participant consent, referring to the study's retrospective design and that the review of medical records did not imply any risk for injury or discomfort of the participants.

All patients had been previously examined with computed tomography of the chest (chest CT), supplemented by positron emission tomography-computed tomography (PET CT) scan when indicated, and had confirmed pulmonary mass and/or mediastinal lymphadenopathy. Subsequently, conventional or advanced flexible bronchoscopies with EBUS-TBNA procedures were performed. We reviewed the outcomes of the pathological assessments of the EBUS-TBNA specimen in all the patients from the study period. To investigate the performance of EBUS-TBNA cytology specimen for histological subtyping of NSCLC and MPS analysis, we included in a consecutive manner all patients where bronchoscopy with EBUS-TBNA revealed metastasis of NSCLC in the examined intrathoracic LN during the study period.

The final patient database included: patient's demographics, clinical characteristics, clinical TNM determined from CT and PET CT scan, the number and location of LN stations sampled, representativity of EBUS-TBNA for every LN position sampled, final histopathological diagnosis, whether mutation/fusion gene status and PD-L1 status were determined on EBUS-TBNA material or other cytology or histology specimen, as well as the portion of bronchoscopies with EBUS-TBNA sampling in which molecular analysis was successful.

## Procedural data

The equipment necessary for the procedure and EBUS-TBNA technique have been fully described by Herth et al. [8] EBUS-TBNA bronchoscopy procedures were performed after topical anesthesia and in monitored moderate conscious sedation with Alfentanil and /or Midazolam commonly in an outpatient setting (Centre for IP in Lund). A convex probe ultrasound flexible video bronchoscope (Olympus BF-UC 180F; Olympus Corp., Tokyo, Japan) was used. Fine needle aspiration was frequently performed with a 22-gauge or sporadically with 21-gauge EBUS-TBNA dedicated needles (Olympus ViziShot, NA-U401SX-4022, and NA-201SX-4022, Olympus Corp.) All the procedures were performed by six operators, four amongst them with many years of experience in the field and two newly trained in the EBUS technique. Each EBUS-TBNA bronchoscopy started with a thorough ultrasound investigation of the intrathoracic LN status. Mediastinal lesions and pathologically enlarged LN, with short-axis larger than 10 mm on CT scan and/or LN with increased metabolic activity on PET CT, were identified and sampled in real-time. Sonographic evaluation of the LN regarding shape, size, margins, echogenicity, presence of hilum, microcalcifications, and necrosis was performed. In concordance with the recommended Guideline for the acquisition and preparation

of EBUS-TBNA specimens for diagnostic and molecular testing, 3–5 passes were made per LN station with 10–15 needle revolutions within the lymph node per pass [14]. Rapid on-site evaluation (ROSE) was used during all EBUS-TBNA procedures. Intraprocedural specimen adequacy was defined by a sufficient number of lymphocytes in the smears and determined in all cases by ROSE. In the post-procedural evaluation of smears and ThinPrep® slides, a representative LN sample exhibited >40 lymphocytes in at least one high-power field, several (5+) clusters of pigmented macrophages, granulomas, or metastasis.

## Predictive analyses

Our department uses a reflex testing approach for lung cancer. Although not mandatory according to guidelines, we aim to perform treatment predictive testing for all advanced NSCLC (also squamous cell carcinomas). The best sample, or samples, from each case, are selected for analysis. Often the cells of a cytological smear are scraped off for MPS while a biopsy is used for diagnostic and predictive IHC, including PD-L1 testing. If no biopsy exists, then a cell block is used for IHC. During Jan 2017 to March 2018, mutations were analyzed with the Ion AmpliSeq™ Colon and Lung Panel v2 (Thermo Fisher Scientific, Waltham, MA) except in cases with a limited amount of DNA when Therascreen® *EGFR* RGQ PCR (Qiagen, Hilden, Germany) was used instead. While *ALK* and *ROS1* fusions were analyzed with IHC for biopsies (clone D5F3, Ventana Medical Systems Inc, Tucson, AZ, and clone D4D6, Cell Signaling Technologies, Leiden, the Netherlands, respectively), FISH was used for cell blocks (Vysis ALK Break Apart FISH Probe and Vysis ROS1 Break Apart FISH Probe, respectively, both Abbott Laboratories, Abbott Park, IL). The reason for this was repeated false-negative *ALK* IHC with the used preparation method for cytology (fixation with CytoLyt® and cell block preparation with Cellient™). FISH was also used when IHC staining was inconclusive and to confirm a positive *ROS1* staining, while FISH confirmation was not mandatory (and hence not always performed) for a positive IHC staining for *ALK* during the study period. From March 2018, mutations and fusions were both analyzed with the Oncomine™ Focus Assay (Thermo Fisher Scientific, Waltham, MA), with PCR for *EGFR* mutations and IHC and FISH for *ALK* and *ROS1* fusions (as prior to March 2018) as backup methods. PD-L1 was assessed with the 28–8 clone (Abcam, Cambridge, UK) until the end of 2017 and with the 22C3 assay (Agilent/pharmDx, Santa Clara, CA) since the beginning of 2018. For both assays, staining was performed on a Ventana Benchmark Ultra (Ventana Medical Systems Inc, Tucson, AZ) using the OptiView visualization system.

Routine mutation, fusion, and PD-L1 analysis at different times during the study period are shown in the S1 File.

## Results

During the study period (January 1, 2017, to April 23, 2018), there were 806 EBUS procedures performed on 765 patients, aiming to diagnose and/or stage suspect or known lung cancer. 296 diagnostic EBUS-TBNA, 260 staging EBUS-TBNA, and 250 radial EBUS procedures were conducted. In patients with concomitant mediastinal lymphadenopathy and lung lesions reachable by radial EBUS, linear EBUS was performed immediately after the radial EBUS, during the same bronchoscopy procedure. The outcomes of the bronchoscopy with radial and/or linear EBUS procedures resulted in diagnosing lung cancer in 209/765 (27%), 174 with NSCLC, 26 with small cell lung cancer (SCLC), and 9 with carcinoids. (Shown in Fig 1).

In 132 of 174 NSCLC patients (76%), we detected cancer cells in the fine needle aspirate from EBUS-TBNA. These patients thus fulfilled the criteria for being included in a consecutive manner in our cohort. Baseline characteristics are shown in Table 1. The majority (95%) of patients were newly diagnosed, whereas relapse or progression of cancer disease and the need

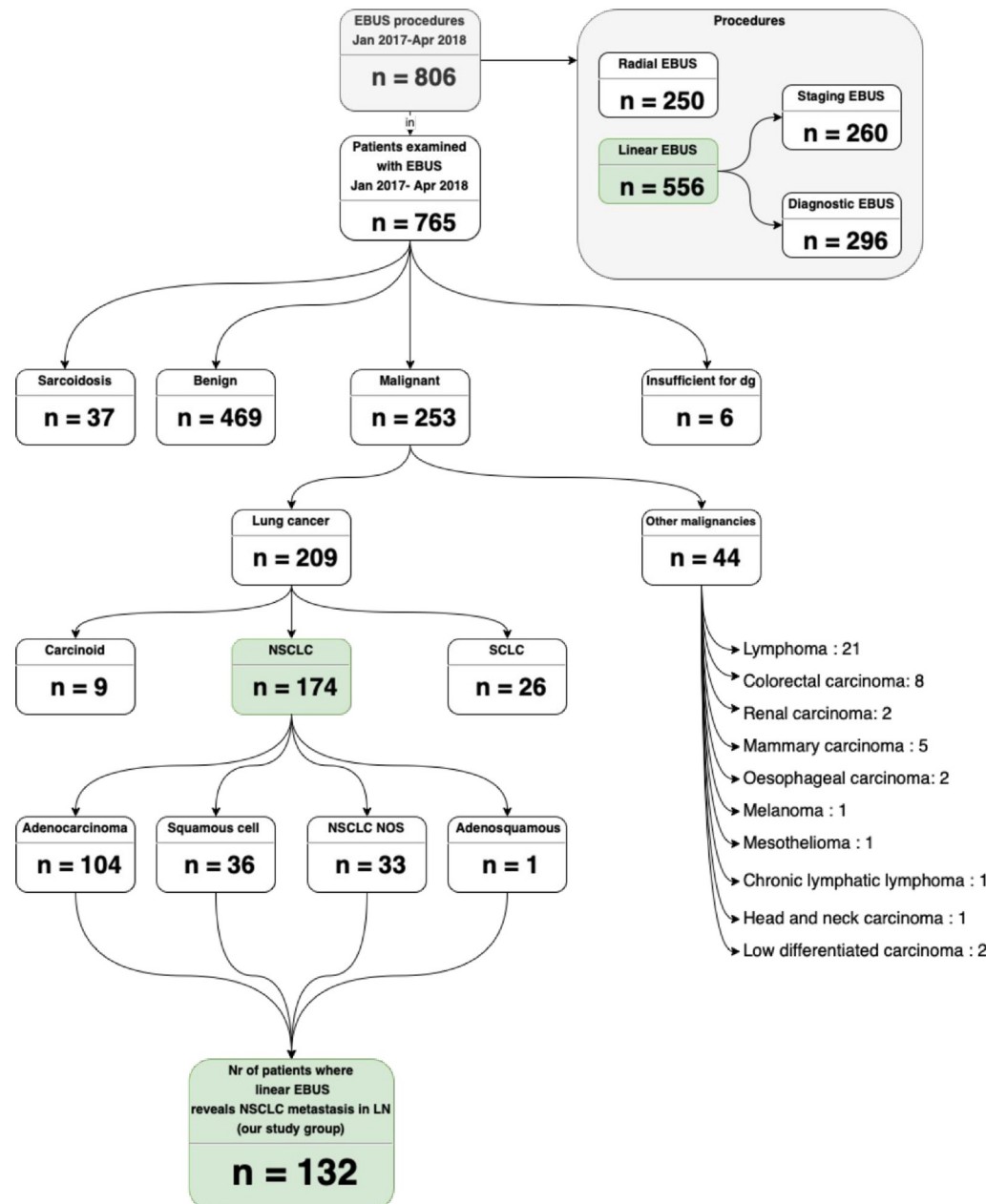

**Fig 1. Flow diagram of examined patients with EBUS-TBNA.** LN—lymph node, NSCLC—Non small cell lung cancer, NSCLC NOS—Non small cell lung cancer not otherwise specified, SCLC—Small cell lung cancer.

for accurate histopathology and molecular status was the reason for EBUS-TBNA in six patients (5%). Initial examination with chest CT and PET CT had classified 63 of 132 patients (48%) in clinical TNM stage III and 61 patients (46%) in clinical TNM stage IV according to the current eighth edition of TNM. The tumors were localized predominantly in the right (35%) or left (17%) upper lobes. In all, 140 bronchoscopies with EBUS were performed in the 132 patients (Diagnostic EBUS bronchoscopy was reattempted in four patients, and two more attempts were made in two patients to establish a diagnosis). The distribution of the tumor

**Table 1. Baseline characteristics of the cohort.**

| Variables | Patients (n = 132) | |
|---|---|---|
| **Sex** | | |
| Male | 63 | 48% |
| Female | 69 | 52% |
| **Age** | | |
| Mean age | 70.1 | |
| **Smoking status** | | |
| Never-smoker | 13 | 10% |
| Ever-smoker | 110 | 83% |
| Not known | 9 | 7% |
| **Comorbidities** | | |
| COPD | 22 | 16% |
| Coronary artery disease | 14 | 10% |
| Diabetes mellitus | 5 | 4% |
| Colorectal cancer | 4 | 3% |
| Breast cancer | 7 | 5% |

COPD—Chronic Obstructive Pulmonary Disease.

lesions according to chest CT and PET CT and sampled LN are shown in Table 2. We have analyzed all histological/cytological slides for the adenocarcinoma cases for subtyping. There was a surgical specimen for 15 of the cases, and for the remaining, there was a biopsy (with/without cytology) in 39 and only cytology in 23 cases. Three cases were mixed mucinous/non-mucinous (one surgical resection and two biopsies), while the other cases were non-mucinous. All three with mixed growth pattern were *EGFR* and *ROS 1* negative, one was *ALK* positive, and one PD-L1 positive (1–4%). Histologic subtypes are reported in Table 3.

In total, 376 LN stations were sampled in 129 of the 132 patients during 140 EBUS-TBNA procedures (in three patients, EBUS-TBNA specimens were collected only from tumor lesions adjacent to the main bronchi). In 32 patients, only one LN was sampled, whereas 2 LN stations were sampled in 28 patients (22%), 3 LN stations in 32 (24%), 4 LN stations in 33 (25%), and 5 LN stations were sampled in 12 patients (9%). The most frequently sampled position was station 7, followed by 4R and 4L. In 126 out of 140 procedures, all sampled LN stations were representative of lymphoid tissue. In 11 procedures (8%), the EBUS-TBNA aspirate was representative from all but 1 LN station.

## NSCLC subtyping and molecular analysis from EBUS-TBNA specimen

In our statistical analysis for the adequacy of the EBUS-TBNA specimen for molecular analysis, we included in a consecutive manner all patients in which the EBUS-TBNA specimen showed cancer cells from NSCLC. We analyzed the outcome of 96 EBUS specimens (out of 140 EBUS bronchoscopies performed in the cohort). We omitted the ones that were performed with the purpose of mediastinal staging and had diagnosis established previously (13 cases), the ones where molecular analysis was attempted on other specimens from the bronchoscopy as bronchial brushes and washes and biopsies (23 cases), and finally those with cell-block from pleural fluid, liver biopsy as well as previous surgery (8 cases).

The choice of the tumor specimen (EBUS cytology, bronchial brush/wash or biopsy) for tumor subtyping and molecular analysis was made at the pathologist's discretion. Final histologic diagnosis and subtyping of NSCLC by IHC on EBUS-TBNA specimen was obtained in

**Table 2. Distribution of tumor lesions and sampled LN.**

| Tumor characteristics | | |
|---|---|---|
| **Tumor location** | | |
| RUL: RML: RLL | 47:06:22 | 35%:5%:17% |
| LUL: LLL | 23:19 | 17%:14% |
| Left hilus | 4 | 3% |
| Right hilus | 6 | 5% |
| Relapse in LN | 3 | 2% |
| Mediastinum | 2 | 2% |
| **T stage** | | |
| T0:T1a:T1b:T1c | 3:2:19:17 | 2%:2%:14%:13% |
| T2a:T2b:T3:T4 | 18:9:31:33 | 14%:7%:23%:25% |
| **Clinical TNM** | | |
| IB | 1 | 1% |
| IIB | 7 | 5% |
| IIIA:IIIB:IIIC | 24:34:05 | 18%:26%:4% |
| IV | 61 | 46% |
| **LN stations sampled by EBUS-TBNA No = 376** | | |
| 12R:11R:10R | 11:43:8 | 3%:11%:2% |
| 4R | 91 | 24% |
| 2R | 8 | 2% |
| 7 | 109 | 29% |
| 4L | 76 | 20% |
| 11L | 32 | 8% |
| 12L:10L:2L:3P | 1:2:1:1 | 1% |
| **Representative LN EBUS/Total No EBUS procedures (140)** | | |
| | **No EBUS** | **% of total EBUS** |
| All representative | 126 (140) | 92% |
| All representative but 4L | 5 (140) | 4% |
| All representative but 4R | 4 (140) | 3% |
| All representative but 11R | 2 (140) | 1% |

RUL—Right Upper Lobe, RML—Right Middle Lobe, RLL—Right Lower Lobe, LUL—Left Upper Lobe, LLL—Left Lower Lobe, TMN—Tumor Node Metastasis, LN—Lymph Node.

66 of 80 executed analyses (83%). Molecular analysis by a panel for MPS was attempted on aspirate from EBUS-TBNA in 96 cases and proved sufficient in 51 (53%). Furthermore, partial genetic analyses (*EGFR* PCR, *ALK* IHC and/or FISH, *ROS1* IHC and/or FISH) were accomplished in an additional 23 out of 96 cases (24%), thus making the EBUS-TBNA aspirate sufficient for analysis of treatment predictive biomarkers in 74/96 cases (77%) of all tested EBUS-TBNA specimens in the cohort.

EBUS-TBNA specimen was insufficient for genetic testing in 22 patients (23%). The EBUS-TBNA specimen showed to be adequate for analysis of *EGFR* gene mutation, *ALK* fusion, *ROS1* rearrangement, and PD-L1 in 69%, 49%, 36%, and 33%, respectively, from the total number of tested EBUS-TBNA samples, as shown in Table 4. The prevalence of *EGFR* gene mutation in the study group was 11% (14/132). We detected exon 19 deletions in five and L858R mutations in seven patients. Two cases harbored *EGFR* exon 20 mutations. *ALK* fusions were identified in four patients (3%), all with lung adenocarcinoma. *ROS1* rearrangement was found in one patient (1%).

**Table 3. Histologic subtype of NSCLC in the cohort.**

| NSCLC histopathologic classification | | | | | |
|---|---|---|---|---|---|
| Lung adenocarcinoma | 77 | | | | 58% |
| | Lung adenocarcinoma non-mucinous | | Lung adenocarcinoma mucinous (mixed) | | |
| | 74 | | 3 | | |
| | Cytology | 22 | Biopsy | 2 | |
| | Biopsy | 35 | | | |
| | Resection | 14 | Resection | 1 | |
| | Biopsy/Resection (metastasis) | 2 | | | |
| Squamous cell lung carcinoma | 33 | | | | 25% |
| NSCLC NOS | 18 | | | | 14% |
| LCNEC (or NSCLC low diff) | 1 | | | | 1% |
| Adenosquamous lung carcinoma | 1 | | | | 1% |
| Sarcomatoid carcinoma | 1 | | | | 1% |
| Lymphoepitelioma-like carcinoma | 1 | | | | 1% |

NSCLC NOS—Non-small cell lung cancer not otherwise specified, LCNEC—Large cell neuroendocrine carcinoma.

**Table 4. Adequacy of the EBUS-TBNA samples for IHC, MPS vs. other tested tumor samples and distribution of predictive biomarkers in the cohort.**

| Oncogen | | IHC for NSCLC subtyping | MPS | EGFR | ALK | ROS1 | PD-L1 | | | |
|---|---|---|---|---|---|---|---|---|---|---|
| | | | | | | | | No of positive (1–49%) | No of positive (50–75%) | No of positive (>75%) |
| No of EBUS procedures (no of patients in the cohort group) | | 140 (132) | | | | | | | | |
| No of EBUS-TBNA samples evaluated for MPS panel | | 96 | | | | | | | | |
| Total No of performed molecular analysis in the cohort (All acquisition methods) | | | | 136 | 125 | 104 | 128 | | | |
| Aquisition method | EBUS-TBNA | 80 | 96 | 96 | 77 | 61 | 69 | 4 | 5 | 4 |
| | Forceps biopsy Bronchoscopy (procedure same time as EBUS) | 28 | 23 | 15 | 23 | 21 | 31 | 6 | 4 | 8 |
| | Bronchial brushings Bronchoscopy (same time as EBUS) | | | 7 | 3 | 4 | 3 | | | |
| | Bronchial washings Bronchoscopy (same time as EBUS) | | | 1 | 1 | 0 | 0 | | | |
| | Pleural fluid Cellblock | 19 | 8 | 1 | 1 | 0 | 0 | | | |
| | Transthoracic biopsies of Lung Liver biopsies and Lymph Node biopsies | | | 11 | 13 | 10 | 15 | 4 | 2 | 3 |
| | Previous or later operation | | | 5 | 7 | 8 | 8 | 2 | 1 | 1 |
| Staging EBUS-TBNA | | 13 | 13 | | | | | | | |
| Method for molecular analysis | MPS | | | 98 | 16 | 17 | | | | |
| | IHC | | | | 32 | 35 | 128 | | | |
| | FISH | | | | 38 | 13 | | | | |
| | PCR | | | 8 | | | | | | |
| | IHC or FISH + MPS | | | | 4 | 1 | | | | |

(*Continued*)

**Table 4.** (Continued)

| Oncogen | | IHC for NSCLC subtyping | MPS | *EGFR* | *ALK* | *ROS1* | PD-L1 | | | |
|---|---|---|---|---|---|---|---|---|---|---|
| | | | | | | | | No of positive (1–49%) | No of positive (50–75%) | No of positive (>75%) |
| **Adequacy** | **No of EBUS specimen with partial genetic analysis** (*EGFR* PCR, *ALK* IHC and/or FISH, *ROS1* IHC and/or FISH) | | 23 (24%) | | | | | | | |
| | **No (%) of adequate from EBUS-TBNA specimen** | 66 (83%) | 51 (53%) | 66 (69%) | 38 (49%) | 22 (36%) | 23 (33%) | | | |
| | No (%) of adequate other methods | 47 (100%) | | 40 (100%) | 48 (100%) | 43 (100%) | 55 (93%) | | | |
| | No (%) of adequate of all analyzed in the cohort (132 patients) | | | 106 (80%) | 86 (69%) | 65 (63%) | 78 (61%) | | | |
| **Positive** | **No of positive from EBUS-TBNA specimen** | | | 10 | 0 | 1 | 13 | 4 | 5 | 4 |
| | No of positive other methods | | | 4 | 4 | 0 | 31 | 12 | 7 | 12 |
| | No (%) of positive of all analyzed in the cohort (132 patients) | | | 14 (11%) | 4 (3%) | 1 (1%) | 44 (34%) | | | |
| **Negative** | **No (%) of negative from EBUS-TBNA specimen** | | | 56 | 38 | 21 | 10 | | | |
| | No of negative other methods | | | 36 | 44 | 43 | 24 | | | |
| | No (%) of negative of all analyzed in the cohort (132 patients) | | | 92 (68%) | 82 (66%) | 64 (62%) | 33 (26%) | | | |
| **Insufficient** | **No (%) of insufficient from EBUS-TBNA specimen** | 14 (17%) | 22 (23%) | 30 (31%) | 39 (51%) | 39 (64%) | 46 (67%) | | | |
| | No (%) of insufficient other methods | | | 0 | 0 (0%) | 0 (0%) | 4 (7%) | | | |
| | No (%) of insufficient of all analyzed in the cohort (132 patients) | | | 30 (22%) | 39 (31%) | 39 (37%) | 50 | | | |

EBUS-TBNA—Endobronchial ultrasound-guided transbronchial needle aspiration, NSCLC—Non small cell lung cancer, IHC—Immunohistochemistry, FISH—Fluorescence in situ hybridization, PCR—Polymerase chain reaction, MPS—Massive parallel sequencing, EGFR—Epidermal growth factor receptor, ALK—Anaplastic lymphoma kinase, ROS1—Proto-oncogene tyrosine-protein kinase ROS, PD-L1—Programmed death-ligand 1.

When molecular analysis on air-dried smears from bronchial brushes or tissue from bronchial biopsies sampled during the same bronchoscopy session as EBUS-TBNA was taken into account, an additional 23 cases had complete molecular analysis. As a result, the cumulative adequacy ratio for complete molecular tumor profiling by MPS from the two procedures performed at the same diagnostic session (flexible bronchoscopy and EBUS-TBNA) was 77%.

Table 4 provides a comprehensive overview of the performed IHC, MPS, and targeted driver oncogenes analyses in the cohort group. It demonstrates the adequacy of the EBUS-TBNA cytology sample for subtyping and genetic testing of NSCLC compared to other cytology and biopsy samples collected in the cohort group.

## NSCLC subtyping and molecular analysis from other tumor specimens in the cohort group

Besides the focus on EBUS-TBNA specimens, we reviewed patient records regarding all other tumor specimens, cytology, and biopsies used for molecular testing to demonstrate the proportion of patients in our cohort group with complete genetic tumor profiling before the

treatment start. A total number of 493 analyses (MPS, *EGFR* PCR, *ALK* IHC and/or FISH, *ROS1* IHC and/or FISH, PD-L1 IHC) of predictive biomarkers were performed on air-dried smears, cellblocks, and biopsy tissue in the study group (all procedures included: EBUS cytology, bronchial brushes and forceps biopsies from bronchoscopy, transthoracic tumor biopsy or tissue from previous or later operation). In 102 and 17 patients out of 132 in our cohort group, a complete or partial molecular analysis was performed, respectively (all procedures included). Molecular analyses were not successful in 11 patients. For two patients with squamous cell lung cancer, the cytology material was not sent to molecular analysis. Additionally, we examined patient records concerning given antitumoral treatment after diagnosis. In 12 out of 14 patients with detected *EGFR* mutation and three out of four patients with detected *ALK* fusion that were classified as stage IV lung cancer disease after diagnosis, has the analysis of treatment predictive biomarkers led to antitumoral treatment with tyrosine kinase inhibitor in the first-line setting. Two patients with locally advanced cancer disease, harboring *EGFR* mutation received concurrent chemoradiation therapy according to the current standard of care. One patient harboring *ALK* fusion was treated with neoadjuvant chemoradiation therapy in a preoperative setting.

The cumulative adequacy and outcome of all tested tumor samples in the cohort for the treatment predictive biomarkers and PD-L1 of two of the cohort subgroups, NSCLC adenocarcinoma, respectively NSCLC NOS, are shown in Table 5.

Complications occurred in 3.6% (5/140) of performed EBUS-TBNA procedures. Postprocedural minor hemoptysis was described in one patient. Two patients reported mild fever a few days after the procedure that resolved spontaneously without any medical intervention. Two patients were hospitalized, one due to sepsis and the other one for mediastinitis, both recovered fully after antibiotic treatment. One patient died 3 days after EBUS-TBNA. The patient was a 77-year-old male with COPD who presented with stage IV lung adenocarcinoma and was admitted to the hospital prior to EBUS-TBNA due to severe dyspnea. The EBUS-TBNA procedure was uncomplicated, but the patient became disoriented the next day. The patient rapidly deteriorated despite the administration of broad-spectrum antibiotics, became unconscious, and died within 72 hours of EBUS-TBNA. The scenario was attributed to sepsis even though the blood culture was negative. A possible differential diagnosis could have been

**Table 5. Adequacy of all tested tumor samples (EBUS cytology, bronchial brush/wash, or biopsy) for analysis of treatment predictive biomarkers in adenocarcinoma and NSCLC NOS, two of the cohort subgroups.**

| | Adequate | | Inadequate | |
|---|---|---|---|---|
| | Adenocarcinoma | NSCLC NOS | Adenocarcinoma | NSCLC NOS |
| *EGFR* | 64 (83%) | 12 (67%) | 13 (17%) | 6 (33%) |
| *ALK* | 55(71%) | 7 (39%) | 22 (29%) | 11 (61%) |
| *ROS1* | 40 (52%) | 8 (44%) | 37 (48%) | 10 (56%) |
| **PD-L1** | 44 (57%) | 7 (39%) | 33 (43%) | 11 (61%) |
| | Positive | | Negative | |
| | Adenocarcinoma | NSCLC NOS | Adenocarcinoma | NSCLC NOS |
| *EGFR* | 12 (16%) | 2 (11%) | 52 (68%) | 10 (56%) |
| *ALK* | 4 (5.0%) | 0 | 51 (66%) | 7 (39%) |
| *ROS1* | 1 (1.0%) | 0 | 39 (51%) | 8 (44%) |
| **PD-L1** ($\geq$1%) | 20 (26%) | 4 (22%) | 24 (31%) | 3 (17%) |

NSCLC NOS—Non-Small Cell Lung Cancer Not Otherwise Specified.

paraneoplastic encephalitis. However, the family strongly opposed an autopsy, and the procedure was not performed.

## Discussion

Present and future novel therapeutic possibilities for lung cancer are persistently challenging the process of genomic profiling of NSCLC, making adequate tissue material for molecular analysis essential. Growing international literature supports the hypothesis that minimally invasive procedures as EBUS-TBNA are sufficient for both diagnosis and genetic testing of NSCLC. However, study results have also shown discrepancies in the diagnostic yield and accuracy of EBUS-TBNA for diagnosis and molecular analysis of NSCLC, thus raising the question of generalizability [20]. In our single-center retrospective study, EBUS bronchoscopy as a part of the initial investigational procedure for suspected lung cancer led to a diagnosis and NSCLC subtyping in 83%, and successful analysis of treatment predictive biomarkers in 77% of all analyzed EBUS samples. Importantly, we also confirm that the majority of EBUS-TBNA cytology specimens can support parallel diagnostics, subtyping and molecular characterization of NSCLC.

### EBUS-TBNA cytology for molecular diagnostics

The EBUS-TBNA cytology samples allowed histological subtyping and successful genomic profiling of NSCLC by MPS concurrently in 53% of all analyzed samples. Partial genomic profiling (PCR *EGFR* and/or FISH/IHC *ALK* and/or *ROS1*) was obtained in an additional 23 cases (24%), seven of which were squamous cell carcinomas where *EGFR* status alone was considered sufficient for treatment decision. Rooper et al. presented similar results where simultaneous subtyping and molecular diagnostics were possible in 57.9% of the patients [19]. EBUS-TBNA specimen was insufficient for genetic testing (by MPS or PCR *EGFR* and/or FISH/IHC ALK and/or ROS1) in 22 of 96 cases (23%) in our study. When reviewing the medical records of those 22 patients aiming to assess if technical issues might have led to poor specimen quality, we could not find anything remarkable other than NSCLC subtyping by IHC has been successfully conducted previously in 8 of total 22 cases (36%) where the EBUS specimen proved insufficient for genetic testing. Several studies have earlier advised moderation in immunohistochemical analysis to preserve tumor specimens to test predictive tumor biomarkers [21, 22].

We further analyzed if there was a correlation between the outcome of the EBUS-TBNA procedure regarding the sufficiency of the EBUS specimen for MPS analysis and the experience of the bronchoscopist. 88 out of 96 EBUS-TBNA procedures were performed by experienced bronchoscopists and 8 by bronchoscopists in training. 47/88 samples obtained by experienced and 4/8 by bronchoscopist in training resulted in an adequate sample for complete MPS analysis. In 21/88 and 2/8, a partial analysis of predictive biomarkers could be performed. 20/88 samples acquired by experienced vs. 2/8 by bronchoscopist in training were insufficient for MPS analysis. The proportion of inadequate EBUS specimens for MPS analysis did not differ significantly between those two subgroups. However, the ratio of procedures performed by inexperienced EBUS bronchoscopists was substantially lower, thus restricting from definite conclusions and requiring further investigation.

### Treatment predictive markers

Fine needle aspirate obtained from LN and lesions adjacent to the main bronchi via EBUS-TBNA was found suitable for the evaluation of *EGFR* (MPS or PCR) in 69% (66/96) against overall adequacy of 80% for all diagnostic procedures in the cohort group. Success

rates for *EGFR* analysis comparable to our study were demonstrated by both Schuurbiers et al. with 77% and by Garcia-Olivé et al. with 72.2% of EBUS cytology samples sufficient for *EGFR* evaluation [23, 24]. Navani et al. presented much higher adequacy, with 90% of EBUS specimen sufficient for *EGFR* analysis [9]. The *EGFR* mutation prevalence of 11% in our cohort is consistent with the findings in several studies analyzed in a systematic review and meta-analysis presented by Labarca et al. [11]. Our results are coherent with the findings by Isaksson et al. who reported an *EGFR* prevalence of 10% in the 519 NSCLC patients subjected to the first 1.5 years of treatment predictive MPS testing in our health care region (2015 to mid- 2016) [25].

The suitability of the EBUS-TBNA cytology specimen in our study for *ALK* analysis by MPS or FISH was 49% (38/77 samples) compared to a 69% adequacy when considering all specimens from every diagnostic procedure conducted in the cohort. The prevalence of *ALK* fusions was 3%, as could be expected from previous studies presented in a systematic review and meta-analysis presented by Labarca et al. [11].

In our study, *ROS1* analysis by IHC/FISH was attempted on 65 EBUS fine needle aspirate specimens. Reliable analysis could be performed in 36% (22/61 samples) presenting *ROS1* positivity in only one case (1%). At present, there are only a few reports on the adequacy ratio of EBUS specimen for *ROS1* and PD-L1 analysis [11]. Fernandes-Bussy et al. conducted a study with 86 cases and found an 83% adequacy ratio for *ROS1* (10/12 samples) [26]. Cicek et al presented a study cohort of 114 patients where *ROS1* analysis could be attempted in 98 cases with a 91% adequacy of EBUS specimens for *ROS1* analysis [27].

Evaluation of PD-L1 expression in lung cancer has mainly been performed on biopsy tissue specimens. The role of EBUS-TBNA in the analysis of PD-L1 is unclear and present evidence insufficient [11]. A very limited number of studies has been conducted, cautiously indicating that an EBUS-TBNA cytology specimen can support PD-L1 testing and quantification and demonstrating satisfactory PD-L1 outcome concordance between cytology and biopsy specimens [28–31]. In our cohort, 128 attempts for PD-L1 evaluation were made on various specimens (cytology, biopsies), showing positive findings (PD-L1 staining in 2–99% of tumor cells) in 44 attempts (35%). However, EBUS-TBNA cytology specimens were used for PD-L1 analysis in only 69 cases, with positive PD-L1 findings in 13 cases. These numbers are lower than seen in large consecutive series and merit further investigation [32, 33].

## Clinical relevance of the study

Our study confirms that diagnostic EBUS-TBNA can be an appropriate procedure for adequate diagnosis, subtyping and genetic testing of NSCLC by MPS. However, the adequacy of the EBUS specimen in our study was lower compared to the outcomes of a few other reported studies [34–37]. Interestingly, in a recent review article, 33 individual studies were analyzed, and a quality assessment found a high risk of bias for selectively reported results in 17/33 studies [11]. According to this report, only two studies were dedicated to the utility of EBUS-TBNA specimen for MPS analysis, both, however, with data insufficient to be included in the meta-analysis [11]. Our approach with unselective inclusion of all assessed EBUS specimens aimed to reflect the everyday reality in most hospitals. We also decided to take an entire consecutive cohort of patients and procedures performed by all our six bronchoscopists with varying levels of expertise, 4/6 with many years of experience in interventional pulmonology whereas 2/6 under training. We believe that this approach of a mixture of expertise levels could allow generalization of our results to the real-world context, which in particular is referring to community hospitals worldwide with a lower volume of EBUS procedures where a considerable quantity of training period is involved [20].

Several previous studies have shown successful MPS on cell block samples collected by EBUS bronchoscopy. A recently published study with a similar cohort size to our study demonstrated that EBUS-TBNA specimen is adequate for genetic profiling of thoracic malignancies by a panel for MPS performed on paraffin-embedded cellblocks in 93% of the cases. The adequacy in the study period improved from 76.3% in the first third of cases to 92.3% for the final one-third of the cases [35]. However, according to our knowledge, our study is one of very few that has in a novel manner (large consecutive cohort) focused on the assessment of large patient series regarding the feasibility of the cytology smears obtained by EBUS-TBNA for molecular cancer profiling by panels for MPS. Our patient material represents the first evaluation of the outcomes from EBUS-TBNA procedures for diagnostics and molecular profiling of NSCLC in a single high-volume academic IP center in Sweden and consists of a genuinely unselective and consecutive series of all patients referred to our IP center for advanced bronchoscopy procedure with EBUS. Importantly, we included in the study all EBUS specimens that showed cancer cells from NSCLC in an indiscriminating manner, aiming to minimize the selection bias.

Our study findings support the benefit of conventional and advanced bronchoscopy combined with EBUS-TBNA performed in one session as a fist-line procedure for prompt and simultaneous diagnosis, precise staging, and genomic profiling of lung cancer. We can also confirm that the analysis of predictive biomarkers is fundamental in process of cancer treatment decision-making. In our patient cohort, the detection of targetable tumor alterations (*EGFR* mutation and *ALK* fusion) has led to treatment with *EGFR/ALK* specific tyrosine kinase inhibitor in all stage IV patients according to the present-day recommendations.

Certain limitations apply to this study. First, we acknowledge the restrictions applicable to retrospective studies. Furthermore, in our real-world approach, we did not re-assess all slides from the EBUS procedures to see if molecular analysis could have also been performed in cases where the clinical pathologist had selected a biopsy for the analysis, nor did we re-evaluate the slides of the group of insufficient ones.

In summary, we believe that the results of our study strengthen the evidence that cytology smears provide a reliable sample with good DNA quality for MPS testing and highlight the need for future studies assessing the suitability of EBUS-TBNA specimen for MPS in a prospective setting.

## Conclusion

Our results confirm bronchoscopy with EBUS-TBNA as a safe, and minimally invasive procedure of first choice with a central role in the demanding diagnostic and staging workout for lung cancer. Although more data is needed on the utility of EBUS-TBNA cytology specimen for MPS and PD-L1 analysis, EBUS cytology aspirates appear to be reliable for diagnosing and subtyping NSCLC and abundantly for treatment predictive molecular testing.

## Supporting information

**S1 File. Routine mutation, fusion, and PD-L1 analysis at different times (the years of inclusion in the study were 01/01/2017-23/04/2018).** Note that for NGS and PCR the best material based on tumor cell content and fraction was selected regardless of specimen type. (DOCX)

**S1 Data. Anonymized study dataset.** Feasibility of EBUS-TBNA for histopathological and molecular diagnostics of NSCLC—a retrospective single-center experience. (XLSX)

## Author Contributions

**Conceptualization:** Marija Karadzovska-Kotevska, Stefan Barath.

**Data curation:** Marija Karadzovska-Kotevska, Jaroslaw Kosieradzki, Lars Ek, Christel Estberg, Stefan Barath.

**Formal analysis:** Marija Karadzovska-Kotevska.

**Investigation:** Marija Karadzovska-Kotevska.

**Methodology:** Marija Karadzovska-Kotevska, Hans Brunnström, Stefan Barath.

**Project administration:** Marija Karadzovska-Kotevska, Maria Planck.

**Resources:** Maria Planck.

**Software:** Marija Karadzovska-Kotevska.

**Supervision:** Stefan Barath, Maria Planck.

**Writing – original draft:** Marija Karadzovska-Kotevska, Hans Brunnström, Maria Planck.

**Writing – review & editing:** Marija Karadzovska-Kotevska, Hans Brunnström, Jaroslaw Kosieradzki, Lars Ek, Johan Staaf, Stefan Barath, Maria Planck.

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
