## [Decision Letter · Decision Letter 0]

25 Oct 2021

PONE-D-21-30444Feasibility of EBUS-TBNA for histopathological and molecular diagnostics of NSCLC - a retrospective single center experiencePLOS ONE

Dear Dr. Karadzovska Kotevska,

Thank you for submitting your manuscript to PLOS ONE. After careful consideration, we feel that it has merit but does not fully meet PLOS ONE’s publication criteria as it currently stands. Therefore, we invite you to submit a revised version of the manuscript that addresses the points raised during the review process.

We look forward to receiving your revised manuscript.

Kind regards,

Andrey Bychkov

Academic Editor

PLOS ONE

Journal Requirements:

3. In the ethics statement in the manuscript and in the online submission form, please provide additional information about the patient records used in your retrospective study, including: a) whether all data were fully anonymized before you accessed them; b) the date range (month and year) during which patients' medical records were accessed; c) the date range (month and year) during which patients whose medical records were selected for this study sought treatment. If the ethics committee waived the need for informed consent, or patients provided informed written consent to have data from their medical records used in research, please include this information.

Additional Editor Comments:

- The section "Predictive analyses" is difficult to read due to a variety of approaches used in different time periods. It would be helpful to add a flowchart on molecular testing (can be placed in a supplement).

- A claim in the conclusion "Our results confirm bronchoscopy with EBUS-TBNA as a safe, cost-effective..." has no actual proof because this particular study didn't address safety or cost analysis. While the latter is definitely out of scope, data on safety (i.e. post-EBUS-TBNA complications) could be assessed by reviewing medical records.

- How many patients in this cohort did receive targeted therapy following molecular analysis? This would probably serve as another proof on "Clinical relevance of the study" (line 390).

Reviewers' comments:

Reviewer's Responses to Questions

5. Review Comments to the Author

Reviewer #1: The research article shows the potential of EBUS-TBNA for histopathological and molecular diagnostics of NSCLC. The manuscript is technically sound. Materials and methods are clearly explained. The data supports the conclusion. I have a few questions for the authors.

1. Is it possible to further subtype adenocarcinoma in cytologic specimens (i.e., mucinous vs non-mucinous) or biopsy/resection specimens (i.e., based on predominant histologic pattern)? If so, the authors may add these subtypes of adenocarcinoma with the results of their corresponding predictive biomarkers in the manuscript.

2. The author stated in Table 3 that NSCLC NOS accounted for 16% of NSCLC in the cohort. NSCLC NOS is a diagnosis of exclusion, in which the tumor does not fit into either adenocarcinoma or squamous cell carcinoma. I think that there are too many NSCLC NOS in this series. The authors should explain in the manuscript how they diagnose NSCLC NOS by morphology and immunophenotype (i.e., TTF1, napsin A, p30, CK5/6, etc.).

Reviewer #2: Authors share their experience of about one year in their center of working-up EBUS samples diagnosed as NSCLC. A total of 132 samples are included in their analysis. This is just a data of single center. No Novelty in the paper. No lessons are learnt from the paper. It appears a narrative of their data. Cytology samples including EBUS is a valuable resource of molecular testing and diagnosis. It has amply been reported in the literature.

Reviewer #3: The main claim of the study is to provide information on effectiveness of the procedure and possible limitations. This study and especially meta-analysis of such studies allow tumor board teams to have balanced decisions when dealing with advanced stage lung cancer.

The manuscript contains detailed description on used protocols and approaches, epidemiology data, cancer stage, applied histological, immunohistochemical and molecular methods with outcomes, which allows to use the study results for meta-analysis.

The manuscript is well organized, written clearly and divided into sections, which allows the reader to easily navigate the text.

The one thing I would like to mention is limited information is provided on failed tests, meaning identify possible reasons for this. Authors stated they reviewed some medical records with no outcome and also mentioned it as a limitation of the study. But, for example, there were both experienced and under training bronchoscopists performing the procedure, but no comparison or data provided to understand if there is a correlation between experience and diagnostic tests outcomes. As it was not a claim of the study, it should be perceived neither a limitation nor a stop sign, but it gives additional value to the data and allows training centers and educational institutions to reflex their approach with field data.

6. PLOS authors have the option to publish the peer review history of their article (what does this mean?). If published, this will include your full peer review and any attached files.

Reviewer #1: No

Reviewer #2: No

Reviewer #3: **Yes: **Borbat Artyom

---

## [Author Response · Author response to Decision Letter 0]

16 Jan 2022

Manuscript PONE-D-21-30444 (Karadzovska-Kotevska et al.)

A detailed point-by-point response to the reviewers’ comments 

We thank the reviewers for their suggestions on how to improve our article "Feasibility of EBUS-TBNA for histopathological and molecular diagnostics of NSCLC - a retrospective single-center experience " and for their constructive comments to further clarify our results. This has helped us improve the manuscript, which we have now revised based on the reviewers’ comments. 

Please find below the detailed point-by-point responses to the reviewers’ comments. Reviewer’s comments are presented in bold font, and our answers are shown in regular text. Text in italics corresponds to the revised version of the manuscript, i.e. after our textual changes, which are also marked in detail as Word Track Changes within the attached new manuscript file ('Revised article with changes highlighted'). The changes are highlighted in yellow for added words or strikethrough for deleted words. Additionally, minor language, grammatical and stylistic errors have been corrected. An unmarked revised version of our manuscript is uploaded as a separate file ('Manuscript').

Journal Requirements:

1. Please ensure that your manuscript meets PLOS ONE’s style requirements, including those for file naming. 

Thank you for this statement. We have now carefully revised our manuscript to comply with the PLOS ONE journal style requirements. Please, inform us if the manuscript should be further updated. 

We thank the reviewers for pointing out that this information was missing in the original manuscript. In the revised version, we have clarified that the ethics committee found no need for participant consent. The following information has been added to the ethics statement in the Materials and methods section:

The regional ethical review board did not request participant consent, referring to the retrospective design of the study and that the review of medical records did not imply any risk for injury or discomfort of the participants. 

3. In the ethics statement in the manuscript and in the online submission form, please provide additional information about the patient records used in your retrospective study, including: a) whether all data were fully anonymized before you accessed them; b) the date range (month and year) during which patients' medical records were accessed; c) the date range (month and year) during which patients whose medical records were selected for this study sought treatment. If the ethics committee waived the need for informed consent or patients provided informed written consent to have data from their medical records used in research, please include this information.

In the revised version of the manuscript, the following information is now included both in the manuscript and in the online submission form:

The Regional Ethical Review Board at the University of Lund, Sweden approved this study (reg nr 2018/730). Marija Karadzovska-Kotevska was permitted by the Ethical Review Board to perform the extraction of patients’ data from the medical records. The data set was fully anonymized for all researchers including Marija Karadzovska-Kotevska before the initiation of the statistical analysis of the material. Patients’ medical journals were accessed between January 2019 and April 2019. Patients included in our study were examined with bronchoscopy with EBUS-TBNA for suspected or known lung cancer at the Unit for Interventional Pulmonology at University Hospital of Skåne in the period between January 2017 and April 2018. The ethical review board did not request participant consent, referring to the retrospective design of the study and that the review of medical records did not imply any risk for injury or discomfort of the participants.

a) If there are ethical or legal restrictions on sharing a de-identified data set, please explain them in detail (e.g., data contain potentially identifying or sensitive patient information) and who has imposed them (e.g., an ethics committee). Please also provide contact information for a data access committee, ethics committee, or other institutional body to 

b) If there are no restrictions, please upload the minimal anonymized data set necessary to replicate your study findings as either Supporting Information files or to a stable, public repository and provide us with the relevant URLs, DOIs, or accession numbers. 

There are no ethical or legal restrictions regarding sharing the raw data for this study. In the revised submission we have prepared a de-identified data set as a Supporting Information file associated with the manuscript.

Answers to additional editor comments:

5. The section "Predictive analyses" is difficult to read due to the variety of approaches used in different time periods. It would be helpful to add a flowchart on molecular testing (can be placed in a supplement).

We appreciate this suggestion from the editor, and we have created the following table, which is accordingly placed in the resubmission as a supplement.

Supplementary Table 1. (Please find the detailed Supplementary Table 1 in the attached "S1 File" and "Response to Reviewers" documents)

6. A claim in the conclusion "Our results confirm bronchoscopy with EBUS-TBNA as a safe, cost-effective..." has no actual proof because this particular study didn't address safety or cost analysis. While the latter is definitely out of scope, data on safety (i.e. post-EBUS-TBNA complications) could be assessed by reviewing medical records.

We thank the reviewer for bringing our attendance to this wording. We have removed parts of this sentence (underlined) from the revised manuscript.

Our results confirm bronchoscopy with EBUS-TBNA as a safe and minimally invasive procedure of first choice with a central role in the demanding diagnostic and staging workout for lung cancer. INSTEAD OF: 

Our results confirm bronchoscopy with EBUS-TBNA as a safe, cost-effective, and minimally invasive procedure of first choice with a central role in the demanding diagnostic and staging workout for lung cancer.

We thank the editor for the good suggestion about safety which led us to perform additional analysis on the safety of the EBUS-TBNA procedure. In the process of data extraction from patients’ medical journals, we collected all documented information on the symptoms and complaints from the patients in the postprocedural time (day 1 to 30). Based on the editors ' remark, we have now performed a detailed analysis of this material and included the following text (in italics) to the Results section of the revised manuscript. 

Complications occurred in 3.6% (5/140) of performed EBUS-TBNA procedures. Postprocedural minor hemoptysis was described in one patient. Two patients reported mild fever a few days after the procedure that resolved spontaneously without any medical intervention. Two patients were hospitalized, one due to sepsis and the other one for mediastinitis. Both recovered fully after antibiotic treatment. One patient died three days after EBUS-TBNA. The patient was a 77-year-old male with COPD who presented with stage IV lung adenocarcinoma and was admitted to the hospital before EBUS-TBNA due to severe dyspnea. The EBUS-TBNA procedure was uncomplicated, but the patient became disoriented the next day. The patient rapidly deteriorated despite the administration of broad-spectrum antibiotics, became unconscious, and died within 72 hours of EBUS-TBNA. The scenario was attributed to sepsis even though the blood culture was negative. A possible differential diagnosis could have been paraneoplastic encephalitis. However, the family vehemently opposed an autopsy, and the procedure was not performed. 

7. How many patients in this cohort did receive targeted therapy following molecular analysis? This would probably serve as another proof on "Clinical relevance of the study" (line 390). 

We appreciate the suggestion from the editor and have now performed an analysis if the assessment of predictive biomarkers has led to specific TKI treatment. We have consequently added the following text (in italics) in the "Results" subheading "NSCLC subtyping and molecular analysis from other tumor specimens in the cohort group” in the revised manuscript. 

Additionally, we examined patient records concerning given antitumoral treatment after diagnosis. In 12 out of 14 patients with detected EGFR mutation and three out of four patients with detected ALK fusion that were classified as stage IV lung cancer disease after diagnosis, has the analysis of treatment predictive biomarkers led to antitumoral treatment with tyrosine kinase inhibitor in the first-line setting. Two patients with locally advanced cancer disease, harboring EGFR mutation received concurrent chemoradiation therapy according to the current standard of care. One patient harboring ALK fusion was treated with neoadjuvant chemoradiation therapy in a preoperative setting.

Based on these findings, we have supplemented the section "Clinical relevance of the study" with the following text (italics) in the revised manuscript.

Our study findings support the benefit of conventional and advanced bronchoscopy combined with EBUS-TBNA performed in one session as a fist-line procedure for prompt and simultaneous diagnosis, precise staging, and genomic profiling of lung cancer. We can also confirm that the analysis of predictive biomarkers is fundamental in cancer treatment decision-making. In our patient cohort, the detection of targetable tumor alterations (EGFR mutation and ALK fusion) has led to treatment with EGFR/ALK specific tyrosine kinase inhibitor in all stage IV patients according to present-day recommendations. 

8. Answers to reviewers comments to the author

Reviewer #1: The research article shows the potential of EBUS-TBNA for histopathological and molecular diagnostics of NSCLC. The manuscript is technically sound. Materials and methods are clearly explained. The data supports the conclusion. 

We thank the reviewer for the careful and positive assessment of our manuscript.

1. Is it possible to further subtype adenocarcinoma in cytologic specimens (i.e., mucinous vs. non-mucinous) or biopsy/resection specimens (i.e., based on predominant histologic pattern)? If so, the authors may add these subtypes of adenocarcinoma with the results of their corresponding predictive biomarkers in the manuscript.

We thank the reviewer for this suggestion. We have now reviewed all histological/cytological slides for patients with lung adenocarcinoma for subtyping and included the following text in the revised manuscript (in italics):

We analyzed all histological/cytological slides for the adenocarcinoma cases for subtyping. There was a surgical specimen for 15 of the cases, and for the remaining, there was a biopsy (with/without cytology) in 39 and only cytology in 23 cases. Three cases were mixed mucinous/non-mucinous (one surgical resection and two biopsies), while the other cases were non-mucinous. All three with mixed growth pattern were EGFR and ROS1 negative, one was ALK-positive, and one PD-L1 positive (1-4%).

The predominant growth pattern for the surgical cases was acinary in seven cases, papillary in three, cribriform in two, solid in one, and mixed mucinous/non-mucinous in one case (as mentioned above). However, six cases had been treated with neoadjuvant therapy, which makes evaluation of predominant growth pattern uncertain. While growth pattern can be assessed in biopsies, it may not be representative of the whole tumor. Given these limitations, we have chosen not to include the data on predominant growth pattern in the manuscript, and we hope the reviewer can agree with this decision. 

2. The author stated in Table 3 that NSCLC NOS accounted for 16% of NSCLC in the cohort. NSCLC NOS is a diagnosis of exclusion, in which the tumor does not fit into either adenocarcinoma or squamous cell carcinoma. I think that there are too many NSCLC NOS in this series. The authors should explain in the manuscript how they diagnose NSCLC NOS by morphology and immunophenotype (i.e., TTF1, napsin A, p30, CK5/6, etc.).

We thank the reviewer for this important comment. The number of NSCLC NOS is quite high. We further reviewed the NSCLC NOS cases (including follow-up samples), and it was possible to re-classify three of them as either adenocarcinoma (AC) or squamous cell carcinoma (SqCC). For the remaining 18, morphology was not conclusive, and the material was either too limited for IHC (n=8) or the case was poorly differentiated with negative markers (n=10). Typically, the cases with sufficient material were stained with TTF-1, napsin A, p40, and CK5 (and often additional markers including CK7, although CK7 was not used to differentiate between AC and SqCC). 

Reviewer #2: Authors share their experience of about one year in their center of working-up EBUS samples diagnosed as NSCLC. A total of 132 samples are included in their analysis. This is just a data of single center. No Novelty in the paper. No lessons are learnt from the paper. It appears a narrative of their data. Cytology samples including EBUS is a valuable resource of molecular testing and diagnosis. It has amply been reported in the literature.

We acknowledge the reviewer’s critical reading and assessment of our manuscript. The data is indeed from a single center which in this case means that the data is population-based from an autonomous health care region in Sweden (where the diagnostic work-up is centralized within each health care region and performed within the public sector). We, therefore, agree that the aim of our study was not primarily to be innovative but rather to concretely determine the value of EBUS-TNA for molecular diagnostics in a large and unselected real-world cohort, aimed to reflect the everyday reality. We believe that large and unbiased studies like ours, to evaluate the suitability of TBNA samples for MPS, have been requested from several previous authors within this field. Moreover, it should be acknowledged that although we (and reviewers 1 and 3) see the study as valuable, the PLOS ONE journal guidelines states that a manuscript should not be rejected based on perceived novelty.

Reviewer #3: The main claim of the study is to provide information on effectiveness of the procedure and possible limitations. This study and especially meta-analysis of such studies allow tumor board teams to have balanced decisions when dealing with advanced stage lung cancer.

The manuscript contains detailed description on used protocols and approaches, epidemiology data, cancer stage, applied histological, immunohistochemical and molecular methods with outcomes, which allows to use the study results for meta-analysis.

The manuscript is well organized, written clearly and divided into sections, which allows the reader to easily navigate the text.

We thank the reviewer for the careful and positive assessment of our manuscript.

The one thing I would like to mention is limited information is provided on failed tests, meaning identify possible reasons for this. Authors stated they reviewed some medical records with no outcome and also mentioned it as a limitation of the study. But, for example, there were both experienced and under training bronchoscopists performing the procedure, but no comparison or data provided to understand if there is a correlation between experience and diagnostic tests outcomes. As it was not a claim of the study, it should be perceived neither a limitation nor a stop sign, but it gives additional value to the data and allows training centers and educational institutions to reflex their approach with field data.

We thank the reviewer for this very important suggestion. We have now performed an analysis on the outcome of the EBUS procedure regarding the sufficiency of the EBUS specimen for MPS analysis and the experience of the bronchoscopist. Our comments on the results of this assessment have been integrated into the "Discussion" section of the revised manuscript in the part with the subheading "EBUS-TBNA cytology for molecular diagnostics (in italics).

We further analyzed if there was a correlation between the poor outcome of the EBUS-TBNA procedure regarding the sufficiency of the EBUS specimen for MPS analysis and the experience of the bronchoscopist. 88 out of 96 EBUS-TBNA procedures were performed by experienced bronchoscopists and 8 by bronchoscopists in training. 47/88 samples obtained by experienced and 4/8 by bronchoscopist in training resulted in an adequate sample for complete MPS analysis. In 21/88 and 2/8, a partial analysis of predictive biomarkers could be performed. 20/88 samples acquired by experienced vs. 2/8 by bronchoscopist in training were insufficient for MPS analysis. A chi-square test of independence showed that there was no significant association between bronchoscopists’ experience and adequacy of EBUS specimen for MPS, X2(1, N=96)=0.03, p=.85. However, the ratio of procedures performed by bronchoscopists in training was substantially lower, thus restricting definite conclusions and requiring further investigation.

---

## [Editor Report · Decision Letter 1]

18 Jan 2022

Feasibility of EBUS-TBNA for histopathological and molecular diagnostics of NSCLC - a retrospective single center experience

PONE-D-21-30444R1

Dear Dr. Karadzovska Kotevska,

We’re pleased to inform you that your manuscript has been judged scientifically suitable for publication and will be formally accepted for publication once it meets all outstanding technical requirements.

Kind regards,

Andrey Bychkov

Academic Editor

PLOS ONE

---

## [Editor Report · Acceptance letter]

24 Jan 2022

PONE-D-21-30444R1 

Feasibility of EBUS-TBNA for histopathological and molecular diagnostics of NSCLC - a retrospective single-center experience 

Dear Dr. Karadzovska-Kotevska:

I'm pleased to inform you that your manuscript has been deemed suitable for publication in PLOS ONE. Congratulations! Your manuscript is now with our production department. 

Kind regards, 

on behalf of

Dr. Andrey Bychkov 

Academic Editor

PLOS ONE